# Secular Trends in Incidence of Esophageal Cancer in Taiwan from 1985 to 2019: An Age-Period-Cohort Analysis

**DOI:** 10.3390/cancers14235844

**Published:** 2022-11-27

**Authors:** Min-Chen Tsai, Yu-Ching Chou, Yu-Kwang Lee, Wan-Lun Hsu, Chin-Sheng Tang, Shiow-Ying Chen, Shih-Pei Huang, Yong-Chen Chen, Jang-Ming Lee

**Affiliations:** 1School of Medicine, College of Medicine, Fu Jen Catholic University, New Taipei City 242008, Taiwan; 2School of Public Health, National Defense Medical Center, Taipei 11490, Taiwan; 3Division of General Surgery, Department of Surgery, National Taiwan University Hospital, Taipei 100225, Taiwan; 4Master Program of Big Data in Biomedicine, College of Medicine, Fu Jen Catholic University, New Taipei City 242008, Taiwan; 5Data Science Center, College of Medicine, Fu Jen Catholic University, New Taipei City 242008, Taiwan; 6Department of Public Health, College of Medicine, Fu Jen Catholic University, New Taipei City 242008, Taiwan; 7Department of Medical Research, Fu Jen Catholic University Hospital, New Taipei City 24352, Taiwan; 8Department of Medical Education & Bioethics, Graduate Institute of Medical Education & Bioethics, National Taiwan University College of Medicine, Taipei 10051, Taiwan; 9Department of Surgery, National Taiwan University Hospital and National Taiwan University College of Medicine, Taipei 100225, Taiwan

**Keywords:** esophageal cancer, squamous cell carcinoma, adenocarcinoma, risk factors, incidence, age-period-cohort

## Abstract

**Simple Summary:**

Esophageal cancer (EC) was the eighth most common type of cancer worldwide in 2020. In Taiwan, the age-standardized incidence of EC, especially esophageal squamous cell carcinoma (ESCC), has increased substantially during the past thirty years. These trends may be associated with changes in the prevalence of risk factors in Taiwan, including smoking, alcohol consumption, and betel nut chewing. In the study, we described the incidence trends of EC from 1985–2019 and the trends of the risk factors to explore the relationship between the risk factors and the incidence rates of EC. The results showed the incidence rate of ESCC in men and overall EC increased prominently from 1985–1989 to 2015–2019. The increased prevalence of risk factors from approximately 1970–1995 could explain the increased cohort effects of EC. We suggest that early detection in high-risk patients and prevention should be conducted strictly.

**Abstract:**

In Taiwan, the age-standardized incidence of EC, especially esophageal squamous cell carcinoma (ESCC), has increased substantially during the past thirty years. We described the incidence trends of EC from 1985–2019 by an average annual percentage change (AAPC) and age-period-cohort model by using Taiwan Cancer Registry data. Age-period-cohort modeling was used to estimate the period and cohort effects of ESCC and esophageal adenocarcinoma (EAC). The Spearman’s correlation coefficient was used to analyze the correlation between age-adjusted incidence rates of EC and the prevalence of risk factors from national surveys. The results showed the incidence rate of ESCC in men (AAPC = 4.2, 95% CI = 3.1–5.4, *p* < 0.001) increased prominently from 1985–1989 to 2015–2019 while that of EAC in men (AAPC = 1.2, 95% CI = 0.9–1.5, *p* < 0.001) and ESCC in women (AAPC = 1.7, 95% CI = 1.4–2.1, *p* < 0.001) increased to a lesser degree. Increased period effects were observed in ESCC in men, ESCC in women, and EAC in men. High correlations were found between the risk factors and the increased birth-cohort effects of ESCC (*p* < 0.05). To conclude, the incidence of ESCC in both sex and EAC in men increased with statistical significance in recent decades. The increased prevalence of risk factors from approximately 1970–1995 could explain the increased cohort effects of ESCC.

## 1. Introduction

EC was the eighth most common type of cancer worldwide [1] and the sixth leading cause of cancer-related death in 2020 [2]. According to GLOBOCAN 2020, the age-standardized incidence rate (ASR) of EC was 9.3 per 100,000 in men and 3.6 per 100,000 in women [2]. Eastern Asia, eastern Africa, and southern Africa have shown the highest ASRs. In Eastern Asia, the ASR was 18.2 per 100,000 in men and 6.8 per 100,000 in women in 2020 [2]. In Taiwan, the ASR of EC has also been high. During the past thirty years, the ASR has increased substantially from 4.88 per 100,000 in 1985 to 23.83 per 100,000 in 2019 in Taiwan.

EC can be histologically divided into two main types: esophageal squamous cell carcinoma (ESCC) and esophageal adenocarcinoma (EAC) [3]. ESCC has long been the most prevalent type in the world. However, there is a difference between developed and less developed countries. EAC has shown an increasing trend and is now the predominant type of EC in the USA and other Western countries [3]. In less developed countries or regions, such as Kenya or Linxian in China, ESCC still has a higher incidence rate [4,5]. These results could be explained by the distinctive risk factor profiles in different countries.

ESCC and EAC each have their own risk factors. Cigarette smoking, alcohol consumption, and betel nut use have been regarded as risk factors for ESCC. For EAC, the risk factors include smoking, obesity, and gastroesophageal reflux disease (GERD) [3,6]. In developed countries, such as the USA, the Netherlands, and the United Kingdom, the growing incidence of EAC is probably due to the increasing prevalence of obesity and GERD [1,3,7]. In less developed countries, the high incidence of smoking, alcohol consumption, hot tea consumption, a poor diet, and betel quid chewing could explain why ESCC is still the predominant type [8].

In Taiwan, ESCC is the major type of EC. From 1979 to 2003, the incidence of ESCC significantly increased, whereas the incidence of EAC leveled off. These trends may be associated with changes in the prevalence of risk factors in Taiwan, including smoking, alcohol consumption, and betel nut chewing [7]. In addition to the traditional risk factors mentioned above, emergent risk factors, such as obesity and *Helicobacter pylori* (*H. pylori)* infection [9], may also influence the trends.

In our study, the aim was to describe the trends of ESCC and EAC from 1985–2019 by an age-period-cohort model and the trends of the risk factors to explore the relationship between the risk factors and the incidence rates of ESCC and EAC.

## 2. Materials and Methods

### 2.1. Study Design and Data Source

We conducted an observational study of patients diagnosed with ESCC and EAC from 1985 through 2019 according to the Taiwan Cancer Registry (TCR) and analyzed the correlation between ESCC, EAC, and their risk factors using data obtained from the Taiwan Tobacco and Liquor Corporation [10] and the Health Promotion Administration [11]. The TCR is a nationwide population-based registration system that provides critical data related to cancer in Taiwan. Every case of cancer must be registered by law [12]. The completeness of the TCR data was 92.8% in 2002 and increased to 99.29% in 2019 [13,14]. The biopsy rates of EC in men and women were 99.47% and 98.45%, respectively, in the 2019 cancer registry annual report. The sex-age-specific population data in Taiwan were obtained from the Department of Household Registration Affairs. The International Classification of Diseases for Oncology Field Trial Edition (ICD-O-FT) and the International Classification of Diseases for Oncology (ICD-O-3) were used to code each histological type in the TCR. The ICD-O-FT codes were used before 2002. The ICD-O-3 codes have been used since 2002. The ICD-O-FT morphology code for esophageal carcinoma is 150. The ICD-O-3 morphology code for esophageal carcinoma is C15. The ICD-O-3 morphology codes for ESCC are 80513, 80523, 80703, 80713, 80723, 80733, 80743, 80753, 80763, 80783, 80833, 80843, 80943, and 81233. The ICD-O-3 morphology codes for EAC are 80453, 81403, 81443, 81453, 82103, 82113, 82553, 82603, 82633, 83803, 84013, 84803, and 84903. The definition of those who smoked and consumed alcohol during 1971–1996 was those over 18 years old who smoked more than three cigarettes per day on average and those over 18 years old who drank more than half a bottle of 0.6 L per month on average, respectively.

### 2.2. Statistical Analysis

The age-specific incidence rates for women and men by calendar period and birth cohort were calculated. The age, calendar period, and birth cohort were divided into 5-year groups, with 18 (0–4, 5–9, 10–14… 85–89), 7 (from 1985–1989, 1990–1994… 2015–2019) and 24 groups (1900–1904, 1905–1909, 1910–1914… 2015–2019), respectively. Age-adjusted incidence rates were calculated using the WHO 2000 standard population. Annual percentage change (APC) and average annual percentage change (AAPC) were conducted through Joinpoint Regression program (Version 4.9.1.0) [15]. 95% confidence interval and *p* values were provided. Age-period-cohort modeling, which was used to estimate the period and cohort effect of ESCC and EAC [16], was conducted using a web tool of the National Cancer Institute. Input data were cases in age groups between 41 and 80, periods between 1985–1989 and 2015–2019 and cohorts between 1905 and 1974. Period and cohort effects are presented as incidence rate ratios. The Spearman’s correlation coefficient was used to analyze the correlation between the prevalence of risk factors, including smoking and alcohol drinking, in 1971–1982 and the incidence of EC, including ESCC and EAC for the period 2006–2017. The time lag between the risk factors and alcohol drinking is considered 35 years [6,17,18,19]. This study was approved by the Institutional Review Board of Fu-Jen Catholic University (C107034).

## 3. Results

From 1985–1989 to 2015–2019, the incidence of EC in men increased dramatically (5.82 to 14.53 per 100,000) with an AAPC of 3.5% (95% CI = 2.3–4.6, *p* < 0.001), while the incidence of EC in women increased slightly but also significantly (0.7 to 0.97 per 100,000) with an AAPC of 0.8% (95% CI = 0.4–1.2, *p* < 0.01) (Table 1 and Figure 1). The incidence rate of EAC in men changed from 0.35 to 0.47 per 100,000, with an AAPC of 1.20% (95% CI = 0.9–1.5, *p* < 0.001). The incidence rate of EAC in women remained relatively stable. (0.06 per 100,000 in 1985–1989 and also 2015–2019, with an AAPC of 0.2% (*p* = 0.56) (Table 1 and Figure 1). The trend of ESCC in men showed a statistically significant increase (4.29 to 13.49 per 100,000) from 1985–1989 to 2015–2019, with an AAPC of 4.20% (95% CI = 3.1–5.4, *p* < 0.001), while the incidence rate of ESCC in women increased to a lesser degree from 1985–1989 to 2015–2019 (0.48 to 0.82 per 100,000), with an AAPC of 1.70% (95% CI = 1.4–2.1, *p* < 0.001) (Table 1 and Figure 1). The numbers and 95% confidence interval of the incidence rates of EC, ESCC, and EAC were shown in Appendix A. 

For different age groups, the incidence of ESCC in men increased as the calendar period progressed (Figure 2a). In the 50–54 age group, the incidence rate was 16.04 in 1985–1989, and it increased to 32.83 in 2005–2009 (*p* < 0.001), which means that the incidence rate in the same age group was higher in later periods. The age-specific incidence rate of ESCC in women remained relatively stable as the calendar period progressed (Figure 2b and Appendix A), except for the age groups 45–49 and 50–54, in which the incidence increases from 0.46 to 1.44 (APC = 5.6, 95% CI = 1.8–9.6, *p* = 0.01) and 0.6 to 2.22 (APC = 5.0, 95% CI = 3.7–6.4, *p* < 0.001), respectively. In the 75–79 age group, the incidence rate was down from 1985–1999 to 2015–2019 with APC was −3.2 (95% CI = −4.6 to −1.8, *p* = 0.01). The incidence rate of ESCC in women increased as age increased. The age-specific incidence rate of EAC in men slowly increased as the calendar period progressed (Figure 2c), in age groups 40–44, 45–49, 50–54, and 60–64 the APC was 3.6 (95% CI = 0.5–6.8), 5.2 (95% CI = 0.3–10.3), 3.5 (1.1–5.9) and 1.6 (95% CI = 0.5–2.8) during 1895–2019, respectively (Appendix A). The incidence rate of EAC in men increased as age increased. The incidence rate of EAC in women remained stable as the calendar period progressed (Figure 2d). The incidence of EAC in women increased as age increased. AAPC and 95% confidence interval in Figure 2a–d was presented in Appendix A.

The incidence of ESCC in men increased as the birth cohort period increased (Figure 3a). The increasing gradients were larger in age groups from 45 to 64 than in the older and younger groups. From age 45 to 49, 1940–1969, the incidence rate increased from 3.32 to 32.04. From age 50 to 54, 1935–1964, the incidence rate increased from 9.17 to 44.51. From age 55 to 59, 1930–1964, the incidence rate increased from 13.82 to 55.62. The incidence rate of ESCC in women remained relatively stable as the birth cohort changed (Figure 3b). However, there was a slight increase in the birth cohort from 1950–1964. In the 45–49, 50–54, and 55–59 age groups, the incidence rate increased from 0.18 to 0.59, 0.21 to 0.40, and 1.51 to 2.22, respectively, during 1950–1964. The incidence of EAC in men remained relatively stable as the birth cohort changed (Figure 3c). An increasing trend was observed between 1935 and 1964. The case numbers were small in the 70–74 and 75–79 age groups, resulting in larger fluctuations in the trends. The incidence of EAC in women remained relatively stable as the birth cohort changed (Figure 3d). The case numbers were so small that larger fluctuations of the trends were observed in the figure.

The results of the age-period-cohort analysis are shown in Figure 4. In ESCC in men, the age effect increased, the rate ratio of the period effect increased from 0.39 to 1.22 from 1985–1989 to 2015–2019, and the cohort effect increased prominently during the birth years 1950–1969. The cohort effect in 1969 was 5.95 times higher than that in 1944 (Figure 4a–c). In ESCC in women, the age effect increased, the rate ratio of the period effect increased from 0.65 to 1.26 from 1985–1989 to 2015–2019, and the cohort effect increased prominently during the birth years 1950–1969. The cohort effect in 1969 was 4.63 times higher than that in 1944 (Figure 4d–f). In EAC in men, the age effect increased, the rate ratio of the period effect increased from 0.60 to 0.97 from 1985–1989 to 2015–2019, and the cohort effect increased during birth years 1950–1974. The cohort effect in 1974 was 4.84 times higher than that in 1944 (Figure 4g–i). In EAC in women, the age effect was increased. The rate ratio of the period effect increased from 0.64 to 1 during 1985–2004 and decreased to 0.46 in 2015–2019. A cohort effect was not observed for EAC in women (Figure 4j–l).

Figure 5 shows a scatter plot of the prevalence of smoking in 1971–1982 and the incidence of ESCC and EAC in men and women in 2006–2017. The Spearman’s correlation coefficients were 0.81 (*p* = 0.02), −0.04 (*p* = 0.93), 0.12 (*p* = 0.77), and −0.45 (*p* = 0.27), for ESCC in men, ESCC in women, EAC in men, and EAC in women, respectively.

Figure 6 shows a scatter plot of the prevalence of alcohol consumption in 1971–1982 and the incidence of ESCC and EAC in men and women in 2006–2017. The Spearman’s correlation coefficients were 0.83 (*p* = 0.02), −0.11 (*p* = 0.80), −0.10 (*p* = 0.82), and 0.20 (*p* = 0.63) for ESCC in men, ESCC in women, EAC in men, and EAC in women, respectively.

Appendix A show the prevalence of smoking and alcohol consumption during 1971–1982 together with the incidence of ESCC and EAC in 2006–2017. The prevalence of smoking in men was high from 1971 to 1990 (58.48% to 59.41%) and then decreased significantly from 1992 to 2016 (57.86% to 28.60%) (Appendix A). The prevalence of smoking in women remained stable from 1971 to 1982 (4.25% to 4.22%), decreased from 1982 to 1986 (4.22% to 2.32%), increased from 1986 to 2002 (2.32% to 5.30%), and then decreased from 2002 to 2016 (5.30% to 3.80%) (Appendix A). Regarding alcohol consumption (Appendix A), the prevalence in men increased from 1971 to 1990 and then decreased from 1990 to 2017 (Appendix A); in women, the prevalence increased from 1971 to 2017 (Appendix A).

## 4. Discussion

By extracting data from the TCR, the 35-year trends of EC, ESCC and EAC, were analyzed in this study. The results show that the incidence rate of ESCC in men increased from 4.29 to 13.49 per 100,000 from 1985–1989 to 2015–2019 with an AAPC of 4.2% (*p* < 0.001), while that of ESCC in women increased from 0.48 to 0.82 per 100,000 with an AAPC of 1.7% (*p* < 0.001). The incidence rate of EAC in men changed from 0.35 to 0.47 per 100,000 from 1985–1989 to 2015–2019, with an AAPC of 1.20% (*p* < 0.001), while that of EAC in women remained relatively stable (0.06–0.06 per 100,000) from 1985–1989 to 2015–2019 with an AAPC of 0.2% (*p* = 0.56). Significant correlations were found between the risk factors and ESCC. The correlation coefficients between the prevalence of smoking and the incidence rates of ESCC and EAC in men were 0.81 (*p* = 0.02) and 0.12 (*p* = 0.77), respectively, and those between the prevalence of alcohol consumption and the incidence rates of ESCC and EAC in men were 0.83 (*p* = 0.02) and −0.10 (*p* = 0.82), respectively.

ESCC accounts for most of cases of EC in the world, with the highest incidence rates in sub-Saharan Africa, central Asia, and east Asia. However, the incidence of ESCC has declined continuously since the 1980s, while that of EAC has increased rapidly [1,20,21,22,23]. In our study, the incidence of ESCC increased from 1985 to 2019, and the peak incidence occurred in younger generations. The possible etiology may be attributed to smoking [24,25], alcohol consumption [24,25,26], and betel nut chewing [27]. The combined effect of these three risk factors accounted for 83.7% of the attributable fraction of EC in ESCC patients [28]. Given that smoking, alcohol consumption, and betel nut chewing are the three main risk factors for ESCC in Taiwan, the increased birth-cohort effect of ESCC may be explained by changes in the prevalence of these risk factors. The prevalence of smoking and alcohol consumption in men were shown in Appendix A, which coincide with the trend of ESCC in men if people started to smoke at about 20 years old [29]. In addition to the birth-cohort effect caused by the risk factors, the high incidence in men may also be explained by a higher prevalence of the ALDH2*2 allele in Han Chinese individuals and a higher level of tobacco use and alcohol consumption [17,30]; additionally, the odds ratio of ESCC has been reported to be higher in people with the ALDH2*2 allele if they have higher levels of tobacco use or alcohol consumption [19]. For women, Tai et al. reported that smoking and alcohol consumption, particularly heavy drinking, are the major risk factors for ESCC in Taiwanese women [18]. The smaller increase in the cohort effect of ESCC in women than men may be due to the following: (1) The prevalence of smoking and alcohol consumption in women was lower than that in men. (2) The odds ratio of smoking was higher in men than in women (men: 4, women: 2.7) [6]. (3) Alcohol intake can interact with tobacco use to increase the risk of ESCC in a multiplicative way [19]. A higher prevalence of smoking and alcohol consumption in men may result in a higher incidence of ESCC in men than that of smoking or alcohol consumption alone. 

The increased birth-cohort effect of EAC was only observed in men from 1950–1974. The major risk factors for EAC are smoking, GERD, and obesity, while *H. pylori* infection is a protective factor. The relative risk of EAC has been found to be 1.85 for ever vs. never smokers [31]. The prevalence of GERD in patients who underwent referral endoscopies increased from 3.4% to 12.4% from 2000 to 2007 [32]. GERD has become a common disorder in Taiwan [33]. The prevalence of obesity has increased significantly in recent decades in Taiwan. In adults, the prevalence of overweight and obesity in men increased from 22.9 to 28.9 and 10.5 to 15.9, respectively, during 1993–2001. However, the prevalence of overweight and obesity in women decreased from 20.3 to 18.7 and 13.7 to 10.7, respectively, during 1993–2001 [34]. The prevalence of *H. pylori* infection increased from 54.4% to 68.3% during 1992–1999 and decreased to 39.2% by 2004. The national policy for the eradication of *H. pylori* and improvement of hygiene and economics may explain its decreased prevalence [4]. According to the trends above, the increased cohort effect in men rather than women may be due to (1) the increased prevalence of smoking in men during 1971–1984 (Appendix A), whereas the prevalence in women had a stable or slightly declining trend during that period (Appendix A), and (2) the increased prevalence of obesity in men but not in women. The increased prevalence of GERD and decreased prevalence of *H. pylori* infection after 1999 may be related to an increasing trend in EAC in the future.

Besides the risk factors mentioned above, the increased trends of ESCC and EAC also partly result from the increased period effect of ESCC in men, ESCC in women, EAC in men, which may be associated with gradual improvements in early detection and endoscopic screening [35], increased convenience of seeking medical care, and increased health awareness, resulting from the National Health Insurance founded in 1995 [5,36]. Similarly, higher incidence of EAC in areas in China may be associated with more health care and early detection, and hence more diagnoses recently [37,38].

The strength of this study lies in the fact that we described the long-term trends of ESCC and EAC in Taiwan from 1985 to 2019 and analyzed them by age-period-cohort modeling. The EC data were extracted from the TCR, which is a population-based registry. As the TCR has a high degree of completeness and accuracy, our data have credibility. In addition, the EC data include the earliest registered cases to the latest cases (1985–2019), which represents the most extensive data set in a publicly accessible database in Taiwan. Although our study lacks individual cancer risk factor data, which impeded direct analyses between the incidence rates of EC and the prevalence of risk factors from the perspective of individuals and may have led to an ecological fallacy, we analyzed the correlation between the long-term trends of risk factors and EC considering a time lag of 35 years. Additionally, the data on the risk factors were from government studies based on population surveys in Taiwan that generally reflect the actual prevalence of the exposure. Other potential confounders were not included in this study, such as obesity, consumption of fresh fruits and vegetables, and good nutritional status, but the importance of smoking, alcohol consumption, and betel nut chewing was greater than that of the other factors.

We suggest that early detection should be conducted strictly [39,40], especially in high-risk patients who have the risk factors mentioned in our study. Prevention is also important, especially in the younger population. Avoiding risk factors, such as by making lifestyle changes, quitting smoking, reducing alcohol and hot beverage intake, and eating more vegetables, should be broadly advocated.

## 5. Conclusions

In conclusion, we described the 35-year trends of EC, including ESCC and EAC. The incidence rate of ESCC in men substantially increased with statistical significance during 1985–2019. In addition, the increase incidence could be seen in ESCC in women and EAC in women. Increased period effects were observed for ESCC in both women and men and EAC in men. An increased birth-cohort effect was observed, which can be explained by the secular trends of their risk factors, such as smoking, alcohol consumption.

## Figures and Tables

**Figure 1 cancers-14-05844-f001:**
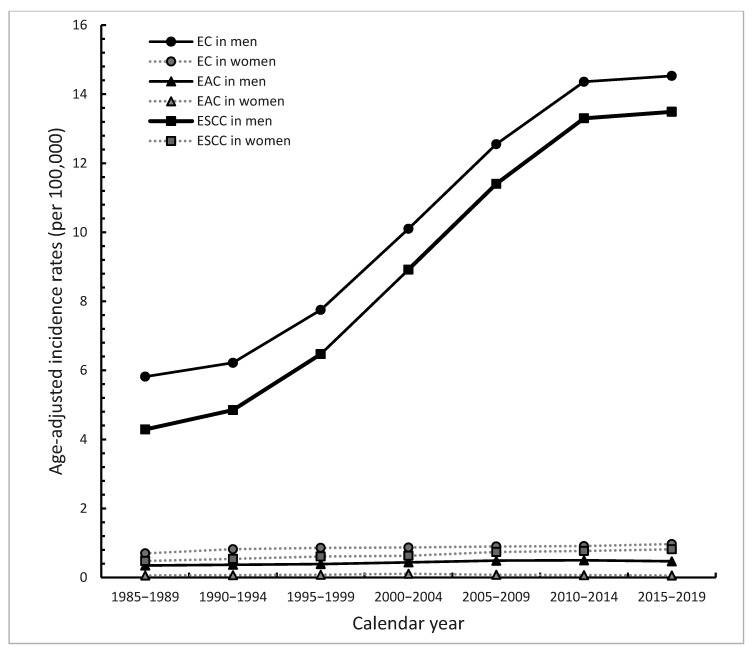
Secular trends of age-adjusted incidence rates of esophageal cancer (EC), esophageal squamous cell carcinoma (ESCC), and esophageal adenocarcinoma (EAC) by sex in Taiwan for the period 1985–2019.

**Figure 2 cancers-14-05844-f002:**
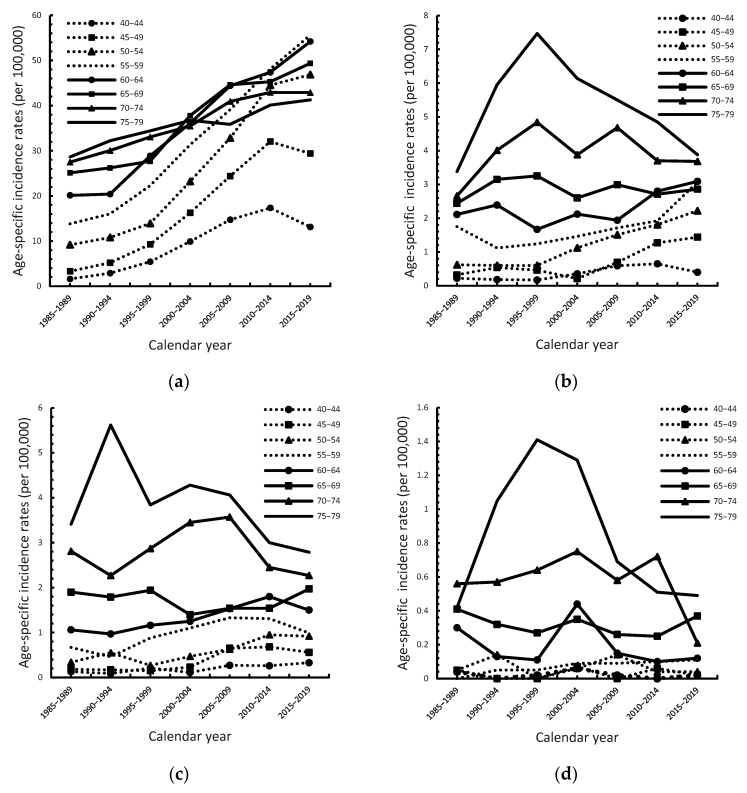
Age-specific incidence rates of esophageal squamous cell carcinoma (ESCC) and adenocarcinoma (EAC) in Taiwan by sex for the period 1985–2019. (**a**) ESCC in men; (**b**) ESCC in women; (**c**) EAC in men; (**d**) EAC in women.

**Figure 3 cancers-14-05844-f003:**
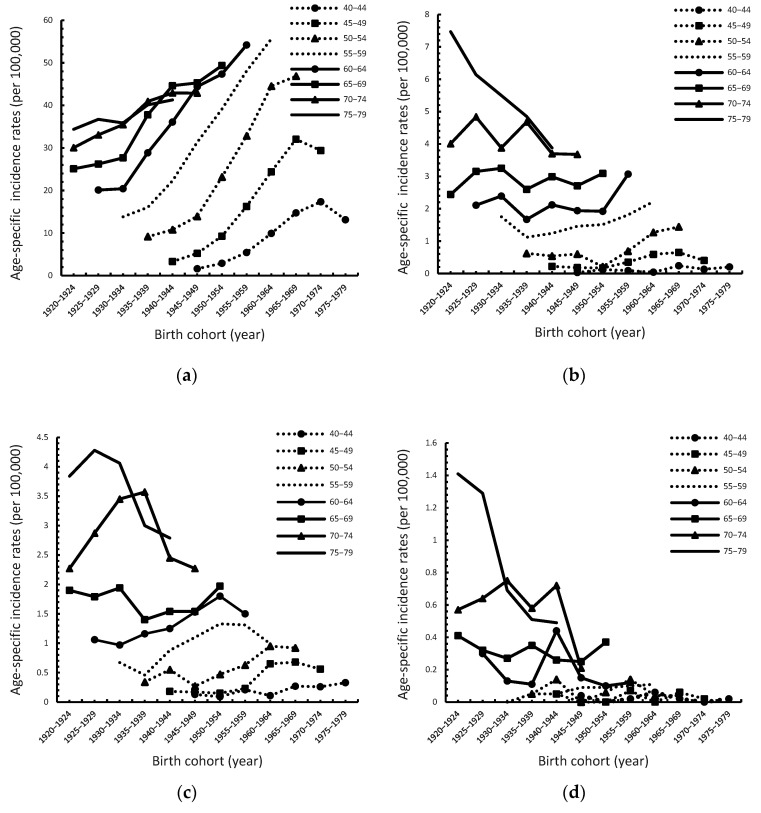
Age-specific incidence rates of esophageal squamous cell carcinoma (ESCC) and adenocarcinoma (EAC) in Taiwan by sex and birth cohort for the period 1920–1979. (**a**) ESCC in men; (**b**) ESCC in women; (**c**) EAC in men; (**d**) EAC in women.

**Figure 4 cancers-14-05844-f004:**
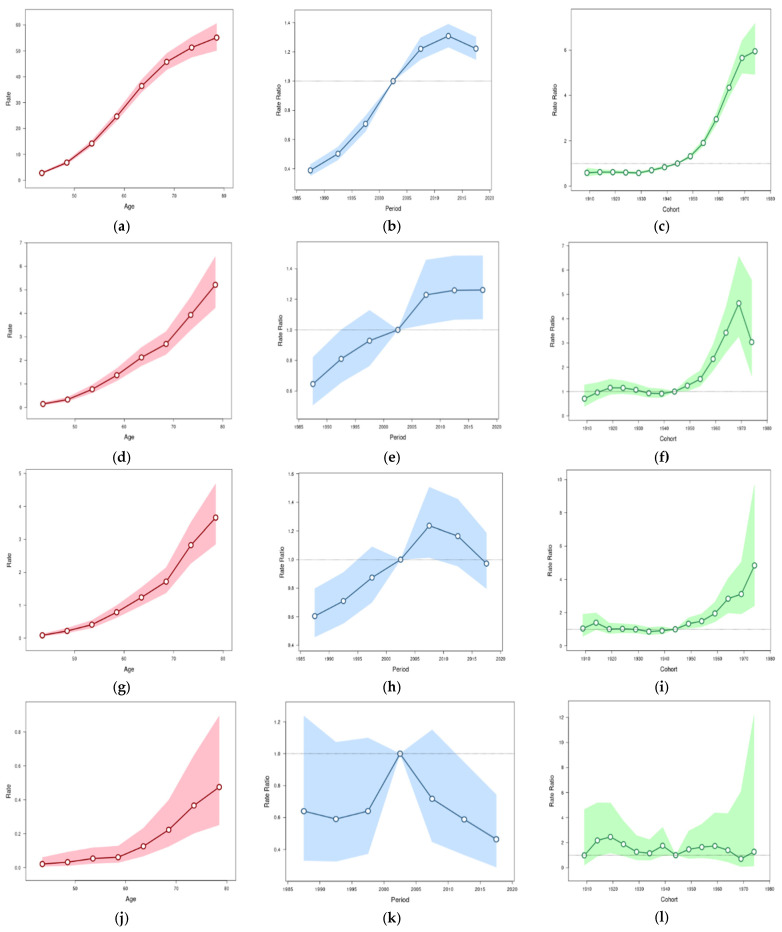
Age, period, and cohort effects of esophageal squamous cell carcinoma (ESCC) and adenocarcinoma (EAC) in Taiwan by sex. Age: 40–80 years old; period: 1985–2019; cohort: 1905–1974. The shadow indicates the 95% confidence interval. (**a**–**c**) Age, period, and cohort effects of ESCC in men; (**d**–**f**) age, period, and cohort effects of ESCC in women; (**g**–**i**) age, period, and cohort effects of EAC in men; (**j**–**l**) age, period, and cohort effects of EAC in women.

**Figure 5 cancers-14-05844-f005:**
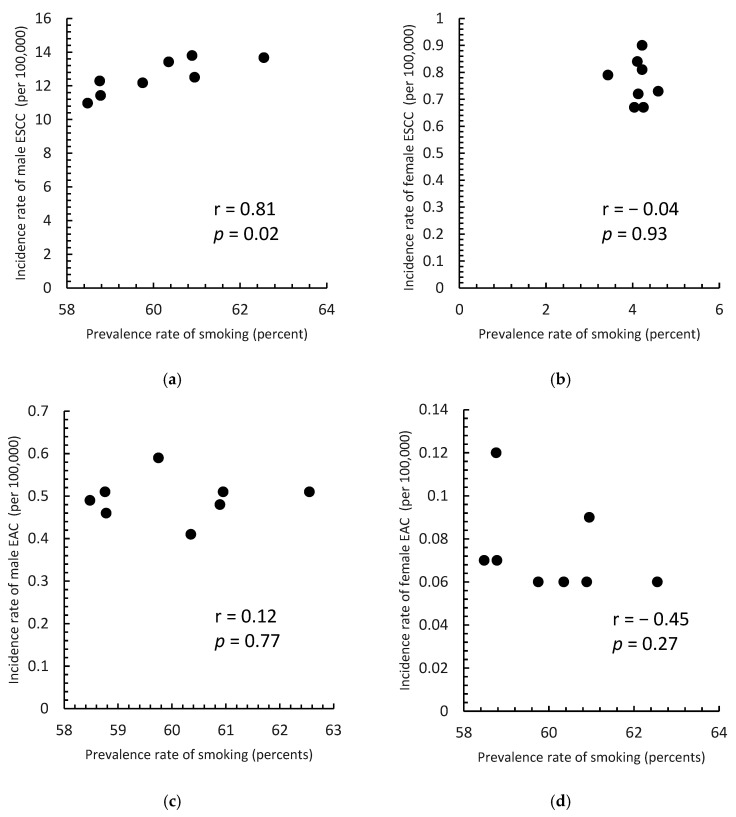
Correlation between the prevalence of smoking in 1971–1982 and the incidence of esophageal squamous cell carcinoma (ESCC) and adenocarcinoma (EAC) by sex for the period 2006–2017. (**a**) ESCC in men; (**b**) ESCC in women; (**c**) EAC in men; (**d**) EAC in women.

**Figure 6 cancers-14-05844-f006:**
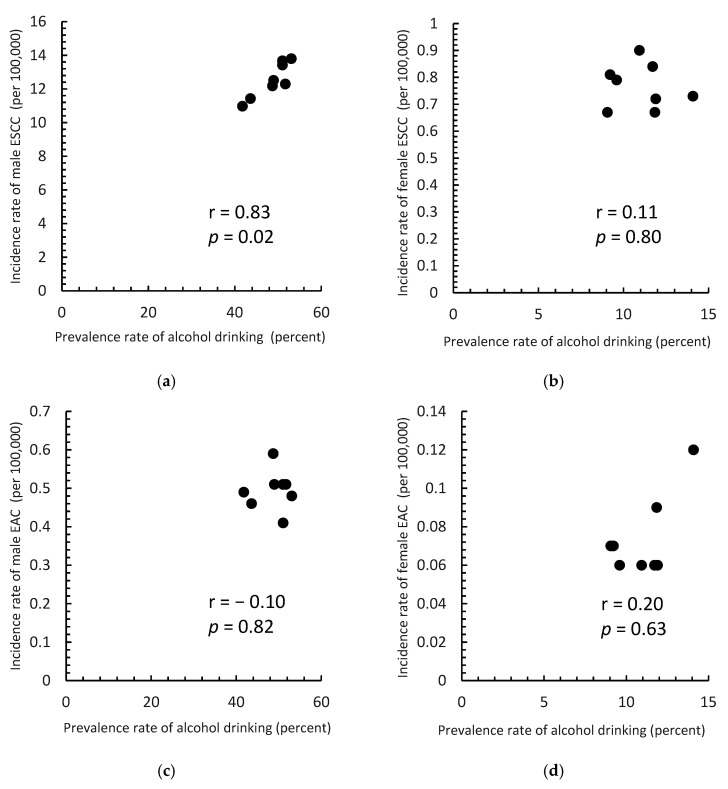
Correlation between the prevalence of alcohol consumption in 1971–1982 and the incidence of esophageal squamous cell carcinoma (ESCC) and adenocarcinoma (EAC) by sex for the period 2006–2017. (**a**) ESCC in men; (**b**) ESCC in women; (**c**) EAC in men; (**d**) EAC in women.

**Table 1 cancers-14-05844-t001:** Annual percent change (APC) and average annual percent change (AAPC) in EC, ESCC and EAC rates for the period 1985–2019, by sex.

Characteristics	Joinpoint Segment Year Start	Joinpoint Segment Year End	APC (95% CI)			*p*-Value	AAPC (95% CI)			*p*-Value
EC										
men	1985	2009	4.4	(1.8~7.2)		0.02	3.5	(2.3~4.6)		<0.001
	2010	2019	1.6	(−3.6~7.1)		0.33				
women	1985	2019	0.8	(0.4~1.2)		<0.01				
ESCC										
men	1985	2009	5.5	(2.9~8.3)		0.01	4.2	(3.1~5.4)		<0.001
	2010	2019	1.7	(−3.1~6.7)		0.27				
women	1985	2019	1.7	(1.4~2.1)		<0.001				
EAC										
men	1985	2009	2.0	(1.3~2.6)		0.01	1.2	(0.9~1.5)		<0.001
	2010	2019	−0.3	(−1.9~1.4)		0.58				
women	1985	2004	4.0	(1.1~7.0)		0.03	0.2	(−0.6~1.1)		0.56
	2005	2019	−3.4	(−5.5~−1.2)		0.02				

## Data Availability

The datasets generated and/or analyzed during the current study are not publicly available in accordance with the policy of the Health and Welfare Data Science Center, Ministry of Health and Welfare, Taiwan, but are available from the corresponding author upon reasonable request.

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
