# Peer review of "Secular Trends in Incidence of Esophageal Cancer in Taiwan from 1985 to 2019: An Age-Period-Cohort Analysis"

_cancers, 2022, doi:10.3390/cancers14235844_

Round 1
Reviewer 1 Report
Secular trends of esophageal cancer in Taiwan from 1985 to 2 2019: An age-period-cohort analysis
Elegant epidemiological study describing changes in cancer incidence over time.
Page 2, “Every case of cancer must be registered by law. These data demonstrate a high degree of completeness and accuracy.” Could you comment on completeness for all the years? I’m thinking particularly of the early period when it was more likely to be incomplete.
Relatedly, Methods, page 3, line102, doesn’t specify whether the same coding version was used for all the years. Please clarify.
Methods, page 3, statistical analysis: “The Pearson correlation coefficient was used to analyze the correlation between the prevalence of risk factors and the incidence of esophageal cancer (Table 1S).” The years of data compared should be included here and a justification of the lag period used.
Same issue, in Results section, since there is no expected r and the sample size is small, it’s hard to judge whether those found “high” are important. Did the authors consider a nonparametric measure?
Page 3, please provide a confidence interval for rates and AAPC, lines 128-136. Ditto for other reports of rates and AAPC throughout.
Discussion: lines 223-225, “The results show that the incidence rate of ESCC in men increased prominently from 4.29 to13.49 per 100,000 from 1985-1989 to 2015-2019, while that of ESCC in women only increased slightly from 0.48 to 0.82 per 100,000.” This statement should be qualified by statistical significance instead of “only increased slightly”. Maybe the authors could consider a percentage change in men and women to emphasize that the growth proportionately is twice as high in men.
Page 10, lines 242-43, assumption that SES is associated. Could this be due to more health care and hence more diagnoses recently? One piece of evidence might be the stage at which the cancer is diagnosed.
Page 12, line372, “lacks individual data” could be qualified to “lacks individual cancer risk factor data”.
Page 13, line 382 (and in the abstract) use of the word “strictly” is not clear as to meaning. Is it according to a guideline? Whose guidance?
Reviewer 2 Report
In my opinion the paper was not clear nor readable, nor satisfactory nor is it informative for anyone requiring a primer to know and understand this issue. Namely, numerous shortcomings in the section, Introduction, Methods, Results and Discussion make this paper not appropriate for publication in this form and significant corrections should be made (major revision). Some comments:
- Lines 56-57: Cite the appropriate reference.
- Lines 57-59: Cite the appropriate reference.
- Lines 55-88: Arrange the section Introduction as a whole so that appropriate references are cited right after stating the information from the literature.
- Lines 55-512: Throughout the entire manuscript correct the references in the text so that they are cited in an appropriate order. The section Introduction cites references No. 37 and 40 after the reference No. 3, etc. Align the order of citing references in the entire text of the manuscript with the list of References.
- Line 112-124: Define the level of significance for all tests used in this manuscript.
- Lines 112-124: State how statistical significance was determined for AAPC. Did you determine the 95%Confidence Intervals?
- Lines 122-124: Table 1S cannot be found in the manuscript nor in the supplementary file. Correct this and explain.
- Lines 126: For the sake of better readability, align the number of decimal spots in the section Results for all numbers, firstly for the values of AAPC.
- Lines 129-136: The stated values are not shown on Figure 1. Add to the paper one new Table that will show values for trends described in the paper.
- Figure 1, Lines 137-140: What does `All men` and `All women` on Figure 1 mean, explain.
- Lines 141-154: Show AAPC for all trends in Figure 2a-d.
- Lines 175-194: Correct and align the markings in the text with markings on Figure 4.
- Lines 195-203: In the text and on Figure 5 insert markings for level of statistical significance.
- Lines 204-210: In the text and on Figure 6 insert markings for level of statistical significance.
- Lines 211-220: What is the main point of Supplementary Figures 1S and 2S? In this paragraph, only an unnecessary long and unuseful description of these two figures was provided. Was it not the aim of the paper to do correlation? Where are the results of the correlation? Correct and explain.
- Lines 254-365: It is very bad practice of writing a paper to describe Figures in this way in the section Discussion. Correct in a way that you discuss the results in the context of results of other studies. Particularly pay attention to statistically significant results (for trends and correlations) in this manuscript. Completely reconstruct the section Discussion.
Round 2
Reviewer 2 Report
Thank you for the opportunity to re-review the manuscript ID: cancers-2004216. The authors have addressed all of the issues highlighted in my review. Thank you to the authors for their responses to my comments. I believe that the changes they have made have significantly improved the manuscript. But, some corrections are need.
Some comments/suggestions:
- In the title, as well as in the entire manuscript, emphasize that it is about `incidence`, e.g. `Secular trends in incidence of esophageal cancer in Taiwan from 1985 to 2019: An age-period-cohort analysis`.
- Since in the revised version the results were corrected with data on statistical significance, statistically significant results should be emphasized throughout the paper (in Abstract, Results, Discussion, Conclusions).
- Lines 45-51: To mach Results in Abstract to the statistical significance of the results presented in the revised version of the manuscript. Correct imprecision and inaccuracies in the Abstract.
Lines 45-47: In the revised manuscript, as well as in the original version of the manuscript (except for abbreviations), it is stated `The results showed the incidence rate of ESCC in men and overall EC increased prominently from 1985-1989 to 2015-2019, while that of ESCC in women increased only slightly.`
But, on Table 1 (Lines 164-166 in revised manuscript) it can be seen that the incidence trend for ESCC in women was continuously increasing during the entire observed period, i.e. by 1.7% per year, with 95%CI (1.4~2.1), and p <0.001, i.e. that a significant growth trend is evident.
To correct.
- Lines 138-139: In section Results was noted `The incidence rate of EAC in men changed from 0.35 to 0.47 per 100,000, with an AAPC of 1.20% (p<0.001).`.
But, in the first paragraph in the Discussion section (Lines: 276-279), it is stated `The incidence rates of EAC in both men and women remained relatively stable (men: 0.35-0.47, women: 0.06-0.06 per 100,000) from 1985-1989 to 2015-2019 (AAPC of EAC in men: 1.2% with p value<0.001, in women: 0.2% with value=0.56).`.
Align Discussion with Results, and correctly cite that the AAPC of EAC for men showed statistical significance of the trend.
- Lines 279-283: In section Discussion was noted `Significant correlations were found between the risk factors and ESCC and EAC. The correlation coefficients between the prevalence of smoking and the incidence rates of ESCC and EAC in men were 0.81 and 0.12, respectively, and those between the prevalence of alcohol consumption and the incidence rates of ESCC and EAC in men were 0.83 and -0.10, respectively.`.
But, in the Results section (Lines: 222-226, as well as on Figure 6), it is stated that a statistically significant correlation was found in men for smoking and and alcohol consumption for ESCC only.
Align Discussion with Results, and correctly cite statistical significance of the correlations.
- Line 527: Since GERD and obesity in this manuscript are not part of the presented results, but are only mentioned in sections Introduction and Discussion, they should not be part of the conclusion of this paper.
Round 3
Reviewer 2 Report
I thank the authors for addressing my comments. The manuscript has been improved accordingly and is now suitable for publication.